# "Table Olive Flours": An Ingredient Rich in Bioactive Compounds?

Nuno Rodrigues [1],*[ID], Catarina Oliveira [1], Susana Casal [2][ID], José Alberto Pereira [1][ID] and Elsa Ramalhosa [1],*[ID]

[1] Centro de Investigação de Montanha (CIMO), Instituto Politécnico de Bragança, Campus Santa Apolónia, 5300-253 Bragança, Portugal; catarinasousa_oliveira@hotmail.com (C.O.); jpereira@ipb.pt (J.A.P.)

[2] LAQV/REQUIMTE, Laboratory of Bromatology and Hydrology, Faculty of Pharmacy, University of Porto, Rua de Jorge Viterbo Ferreira 228, 4050-313 Porto, Portugal; sucasal@ff.up.pt

\* Correspondence: nunorodrigues@ipb.pt (N.R.); elsa@ipb.pt (E.R.); Tel.: +351-273303277 (N.R.); +351-273303308 (E.R.)

**Featured Application: This ingredient will use the surplus of table olive fruits that do not have the characteristics to be marketed, such as fruits with no commercial size or containing some visual defects. In this way, these fruits can be valued through flour production. The production of this new product will also reduce waste and be more sustainable, in line with the objectives of the circular economy.**

**Abstract:** The aim of this study was to produce different "table olive flours" from fruits at different maturation stages. "Table olive flour" is here presented as an innovative product that can gain importance as a bioactive rich ingredient. Three types of natural table olives from cv. Cobrançosa, i.e., green, turning color, and black olives, were soaked, freeze-dried and ground to obtain three different flours. Their physical and nutritional characterization, lipid fraction (fatty acids and tocopherols profiles), phenolic profile, and antioxidant activity (total reducing capacity, radical scavenging activities of DPPH and ABTS$^{\bullet+}$) were analyzed. "Table olive flours" with different colors and different characteristics were obtained. The "green table olive flour" had the lowest fat content and energy. On the contrary, it showed the highest protein, carbohydrate contents, percentages of oleic acid (C18:1), and MUFA, as well as total tocopherols. It also showed the highest antioxidant activity. The "black table olive flour" was the one with the highest percentages of palmitic acid (C16:0), SFA, and total reducing capacity. In the three types of developed "flours", nine phenolic compounds were detected, hydroxy-tyrosol being the major, followed by tyrosol, and luteolin. In conclusion, from natural table olives of cv. Cobrançosa, different "table olive flours" with distinct properties and high amounts of health-promoting compounds can be produced.

**Keywords:** table olive flours; fatty acids; tocopherols; phenolic compounds; antioxidant activity

## 1. Introduction

Consumer preferences, expectations, and dietary patterns are a constant challenge to food industries [1]. Consumers are seeking uniqueness, and the use of traditional products in new presentations and applications to fully value these products is a great possibility. There is a concern to solve nutritional issues such as dietary imbalances (lack of specific nutrients, antioxidants, and vitamins), and the needs of consumers of new ingredients which are practical and easy-to-use that also meet environmental and sustainability concerns. Thus, more innovations are needed, particularly in the direction of plant-based products that are healthy, as well as towards the use of clean labels [2]. Consumers recognize plant-based diets as ethical, healthy, and environmentally friendly [2]. Furthermore, fruit and vegetable by-products may be transformed into fiber-rich flours and bioactive compounds [3]. Consumers also look for convenient, tasty plant-based products based on simple ingredient lists [2]. Table olives are included in this group, recognized as having

high nutritional value, healthy fat, and a considerable amount of antioxidants. Some preparations of table olives are recognized to be more beneficial to human health than others. The natural table olives obtained by natural fermentation [4] do not use additives other than salt. However, using high sodium chloride levels as a preservative has been one of the negative aspects which has been highlighted as limiting their consumption. Some reviews have revealed that the applied processing method influences the nutritional properties of table olives; nevertheless, some preharvest factors such as irrigation and fruit ripening stage may have a certain weight. The nutritional value of table olives depends mainly on the balanced profile of polyunsaturated and monounsaturated fatty acids, as well as the contents of health-promoting phenolic compounds [5,6]. Thus, new forms of presentation must be considered. It is also necessary to make table olives production more sustainable, in addition to finding ways to value smaller fruits and/or olives with small defects, which do not allow them to be marketed in that way. The production of "table olive flour" can be an interesting method to obtain a product that can be added to other dishes or products (such as sauces) to enrich their bioactive properties. Furthermore, a product with lower salt content than the fresh olives can be obtained since, before producing the flour, the fruits can be soaked to remove a large part of the salt present in the fruit.

Likewise, the olive fruits' phenolic composition is well studied and documented [4,7–12], bringing benefits to human health. Several phenolic compounds such as hydroxytyrosol, tyrosol, rutin, luteolin-7-*O*-glucoside, verbascoside, oleuropein, comselogoside, and ligstroside have been reported in olives. Concerning table olives, those produced through natural fermentation are the most prominent, as processes involving oxidation steps such as the Californian style may cause a significant reduction in the phenolic contents [4]. It is also known that the maturation index influences the antioxidant content [11], and the applied technological processes to produce edible table olives can also influence the final product [4,8,12]. The development of dried products based on table olives will allow for their consumption. So far, some studies on table olives drying by hot air have been carried out [13–19]. Temperatures at 40 [19], 45 [14], and 50 °C [13,15,18], as well as ranges between 40–70 °C [16,17], have been applied; however, only in the studies performed by Borzillo et al. [14], Lanza et al. [18] and Mantzouridou and Tsimidou [19] were the total phenols or individual phenols analyzed. In general, the hot air drying caused a decrease in phenol levels. Freeze-drying may be a good alternative to hot air drying. This technology has been applied to olives, preserving antioxidants [12,20], but it has not yet been explored for drying table olives. Considering that table olives are a nutritionally rich food that can be used in different applications, new products must be developed to meet new audiences and different uses. Thus, the present work intends to develop "table olive flours". This healthy and innovative product, prepared with different types of natural table olives (green, turning colors, and black), was characterized concerning the physical-chemical and antioxidant properties of table olives. Furthermore, in this way, the waste generated in the production of olives will be reduced, in line with the principles of sustainability and circular economy.

## 2. Materials and Methods

### 2.1. Sampling

Table olive samples obtained through natural fermentation of cv. Cobrançosa were purchased from a local producer from Trás-os-Montes (Mirandela region, Portugal). In total, three types were selected (green, turning colors, and black table olives), corresponding to different maturation stages used for table olives processing. According to the producer, raw fruits proceeded from the same olive grove and were subjected to a similar natural fermentation process. For the production of table olives by this process, the fruits were placed in a fermenter with a 7% brine (NaCl) solution, and the natural fermentation was allowed to take place, which lasted approximately five months until they became edible. During the process, the pH and temperature were controlled. For each table olive type (green, turning color, and black), three buckets of 5 kg were obtained. In the laboratory, in order to remove salt, table olives were placed in distilled water for 48 h, with the water

changed only once after 24 h. Then, the stones were removed, and the samples were frozen at −21 °C, freeze-dried, and milled until the flour was obtained. The obtained flours (three per type of table olive) were placed in amber containers until the first tests were carried out.

Each independent table olive flour (n = 3 per type) was then subjected to different analyses, and each parameter was analyzed in triplicate.

### 2.2. Colour

Flour color was determined with the Konica Minolta CR-400 colorimeter (Osaka, Japan), using the CIELAB scale, evaluating the coordinates *L\**, *a\**, *b\**, *C\**, and *h*, where *L\** represents the luminosity and varies between 0 (black) and 100 (white), coordinate *a\** extends from green (−*a\**) to red (+*a\**), and coordinate *b\** extends from blue (−*b\**) to yellow (+*b\**). The *C\** indicates the purity or intensity of the color, and the *h* is the hue.

### 2.3. Nutritional Composition

The nutritional composition of the different flours was determined according to the AOAC Official Methods [21], including moisture (925.40), total extractable fat (948.22), using petroleum ether for a minimum extraction time of 24 h, protein content (920.152), and total ashes (940.26). The carbohydrate plus fiber content was estimated by difference. The energy value expressed in kcal/100 g of dry matter was calculated following the Atwater system, using a factor of four for protein and carbohydrates, and nine for extracted lipids. The moisture content was also determined in the table olives used to produce the flours.

### 2.4. Characterization of the Lipid Fraction

For characterization purposes, fat was extracted from the flours in a Soxhlet device for 6 h, using petroleum ether as a solvent enriched with 0.01% BHT (di-*tert*-butyl methyl phenol). This extracted fat was further characterized for its fatty acid and tocopherol profiles, as described in the following sections.

#### 2.4.1. Fatty Acids Composition

Fatty acids were assessed as methyl esters, after cold alkaline transesterification with methanolic potassium hydroxide solution, following the Commission Regulation (ECC) n∘ 2568/91 [22]. The fatty acid profile was established using a Chrompack CP 9001 chromatograph, a split-splitless injector, an FID detector, an autosampler Chrompack CP-9050, and a fused silica Select FAME capillary column (50 m × 0.25 mm i.d.; Varian, Palo Alto, CA, USA). The carrier gas was helium at an internal pressure of 140 kPa. The detector temperature was 270 °C, and the injector was kept at 250 °C. A 1:50 split ratio was used, with 1 µL injected. The fatty acid contents were quantified in relative percentage, calculated by internal normalization of the chromatographic peak area eluting between myristic and lignoceric methyl esters. A fatty acids methyl esters standard mixture (Supelco 37 FAME Mix, Bellefonte, PA, USA) was used for identification and calibration purposes (Sigma, Madrid, Spain).

#### 2.4.2. Tocopherols Composition

Tocopherols were assessed following the ISO 9936 method [23], with some modifications [24]. Tocopherol standards (α-, β- and γ-) were purchased from Sigma (Spain), and 2-methyl-2-(4,8,12-trimethyltridecyl) chroman-6-ol (tocol), used as internal standard, was from Matreya Inc. (Pleasant Gap, PA, USA). An accurate amount of extracted lipids (50 mg), with internal standard solution (tocol, 100 µg/mL prepared with n-hexane), were diluted in hexane, mixed, and then centrifuged for 5 min at 13,000 rpm, with the obtained supernatant analyzed by high-performance liquid chromatography (HPLC). A Jasco integrated system (Tokyo, Japan) was used, comprising a Jasco LC-NetII/ADC data unit, a PU-1580 Intelligent Pump, a MD-4020 photodiode array detector, and an FP-4020 fluorescence detector (λexc = 290 nm and λem = 330 nm, gain 10). For the chromatographic separation, a Luna silica column (3 µm) 100 × 3.0 mm (Phenomenex, Alcobendas, Spain), with the respective

guard column, at 23 °C was used. The eluent was a mixture of n-hexane and 1,4-dioxane (97.5:2.5), at a 0.7 mL/min flow rate. Data were analyzed with the ChromNAV Control Center—JASCO Chromatography Data Station (Japan). The compounds were identified using standards, considering the co-elution retention time, and according to their UV spectra. Quantification was based on the internal standard method, using the fluorescence signal response and individual calibration curves for each tocopherol. The total vitamin E was quantified as the sum of the individual tocopherol contents.

### 2.5. Antioxidant Activity

#### 2.5.1. Preparation of Phenolic Extracts

Extracts were prepared from olive flours (1.5 g) to which 50 mL of methanol was added. After stirring for 60 min at room temperature, the mixture was filtered through Whatman No. 4 filter paper. After this, two further extractions were performed in the solid remains, each for one hour under stirring. Finally, methanol was evaporated from the combined extracts using a rotary evaporator (Stuart RE 3000) at 35 °C. The dried extracts were weighed and dissolved in methanol to obtain a known extract concentration (50 mg extract/mL). This solution was used for the phenolic profile, total reducing capacity, DPPH and ABTS$^{\bullet+}$ radical scavenging effects, as detailed below.

#### 2.5.2. HPLC-DAD Phenolic Profile

The phenolic profile of table olive flours was determined by high-performance liquid chromatography (HPLC) in a Knauer SmartLine separation module, equipped with an automatic injection system (autosampler 3800), at 4 °C, and with a photodiode array detector (PDA). The data were obtained using the ClarityChrom® software. To achieve the separation of the compounds, a reverse phase C18 Nucleosil (Macherey-Nagel) column (Spherisorb ODS2), $250 \times 4$ mm id (5 μm), was used, and maintained at 30 °C (through the heating oven). The separation was carried out using a gradient system composed of water/formic acid (19:1) (A) and methanol (B). The flow was 0.9 mL/min with the following gradient: 5% B at 0 min, 15% B at 3 min, 25% B at 13 min, 30% B at 25 min, 35% B at 35 min, 40% B at 39 min, 45% B at 42 min, 45% B at 45 min, 47% B at 50 min, 48% B at 60 min, 50% B at 64 min, and 100% B at 66 min. The chromatographic data were recorded at 280 nm and 330 nm. The spectral data of all the peaks were accumulated in the 200–400 nm range. The identification of the phenolic compounds was carried out by comparing the retention times and the spectra of the chromatographic peaks with those of standards analyzed under the same conditions. The quantification of phenolic compounds was carried out by recording the absorbances in the chromatograms and calibration lines prepared from external standards. Pure standards of hydroxytyrosol, tyrosol, chlorogenic acid, verbascoside, rutin, luteolin, and apigenin were acquired from Extrasynthese.

#### 2.5.3. Total Reducing Capacity

The spectrophotometric method described by Singleton and Rossi [25] was used to determine the total reducing capacity, with some modifications.

In a test tube, 1 mL of the extract solution and 1 mL of the Folin-Ciocalteu solution were mixed and left to stand for 3 min. Simultaneously, a blank was prepared with the sample replaced by methanol. Then, 1 mL of a saturated sodium carbonate solution ($Na_2CO_3$) and 7 mL of distilled water were added. After vortexing, the solution was left to stand in the dark for 1 h and 30 min, and the absorbances were read on the UV-visible spectrophotometer at a wavelength of 725 nm. Each test was performed in triplicate. The calibration curve was prepared with gallic acid, and the results were expressed in mg of gallic acid equivalents (GAE)/g extract.

#### 2.5.4. DPPH Radical Scavenging Effect

The antiradical activity of table olive flours was determined using the 2,2-diphenyl-1-picrylhydrazyl (DPPH) free radical blocking method, according to the methodology

described by Hatano et al. [26]. The determination of the blocking capacity of DPPH free radicals consisted of mixing 0.3 mL of sample extract (0.3 mL of methanol was used for the blank) and 2.7 mL of a methanolic solution containing DPPH radicals ($6\times10^{-5}$ mol/L). After vortexing, it was placed for one hour in the dark at room temperature, and read at 517 nm on the UV-visible spectrophotometer. The DPPH radical scavenging effect was calculated as the percentage of DPPH discoloration using the following equation:

% DPPH radical scavenging capacity = $[(A_{DPPH} - A_S)/A_{DPPH}] \times 100$, where a$_S$ is the absorbance of the solution when the sample extract was added, and $A_{DPPH}$ is the absorbance of the DPPH solution.

### 2.5.5. ABTS$^{\bullet+}$ Radical Scavenging Effect

The total antioxidant capacity of table olive flours was additionally tested by evaluating the sequestering activity of the radical ABTS$^{\bullet+}$ (radical 2,2′-azino-*bis*-3-ethylbenzothiazoline-6-sulfonic acid), according to the methodology described by Sánchez et al. [27]. To carry out this method, and after preparing the ABTS$^{\bullet+}$ solution calibrated at $0.700 \pm 0.020$ at 734 nm, 2 mL were used for each 100 μL of the extract solution of table olive flour (2 mg of extract/mL), where, after vortexing, the reaction took place for 6 min. The absorbance was determined at 734 nm.

The ABTS$^{\bullet+}$ radical scavenging effect was calculated as the percentage of ABTS$^{\bullet+}$ discoloration using the following equation:

% ABTS$^{\bullet+}$ radical scavenging capacity = $[(A_{ABTS} - A_S)/A_{ABTS}] \times 100$, where $A_S$ is the absorbance of the solution when the sample extract was added, and $A_{ABTS}$ is the absorbance of the ABTS$^{\bullet+}$ solution.

### 2.6. Statistical Analysis

The program used in the statistical analysis was Minitab (version 14, Minitab Ltd., Coventry, UK). We started by assessing the normality and homogeneity of the variances using the Shapiro–Wilk and Levene tests, respectively. The data were found to be normal. When homogeneity of variances was observed, ANOVA was applied. Then, in case of significant differences between samples ($p < 0.05$), the Tukey test was applied. In situations where no homogeneity of variances was observed, ANOVA–Welch was used to detect significant differences between samples. In the case in which this occurred, the Games–Howell test was then applied. A principal component analysis (PCA) was also performed on the results of the three table olive flours. The PCA score plot was used to differentiate the table olive flours and verify the role of table olives (green, turning color, and black) on their properties, namely color, fatty acid and tocopherol compositions, individual phenolic compounds, and antioxidant activity. The number of components to retain for data analysis was evaluated by: (i) the respective eigenvalues (must be >1); (ii) Cronbach's $\alpha$ parameter (that should be positive); and (iii) the total percentage of variance (that should be as high as possible), explained by the number of components selected.

## 3. Results

### 3.1. Colour

Food color is one of the first parameters to be perceived by the consumer, and is very important to dictate a product's success and uses. Table 1 shows the mean values for the color parameters measured in the CIELAB color space for the three table olive flours. Significant differences ($p < 0.05$) were found between them. The *L\** coordinate varied between 35.8 and 49.2, with the highest value obtained for the "green table olive flour", indicating that this sample had higher luminosity than the other flours. The "black table olive flour" showed a lower luminosity, since its color approached black.

Regarding the *a\** coordinate, the "black table olive flour" was the flour with the highest *a\** value (11.6), indicative of a more reddish hue than the others. In contrast, the "black table olive flour" presented a lower *b\** reading ($p < 0.05$) (5.4), suggesting a less yellowish hue, while the "green table olive flour" had the highest *b\** value (24.9).

**Table 1.** Color parameters and nutritional composition (grams per 100 g dry matter) and energy value (kcal per 100 g dry matter) of table olive flours (mean ± standard deviation).

| Physical Parameters: | Table Olive Flour | | | |
| --- | --- | --- | --- | --- |
| | Green | Turning Colour | Black | *p*-Value |
| *L** | 49.2 ± 5.5 [c] | 43.7 ± 4.3 [b] | 35.8 ± 1.5 [a] | <0.001 |
| *a** | 3.6 ± 0.6 [a] | 6.3 ± 0.1 [b] | 11.6 ± 0.4 [c] | <0.001 |
| *b** | 24.9 ± 2.8 [c] | 16.7 ± 2.6 [b] | 5.4 ± 1.0 [a] | <0.001 |
| *C** | 25.2 ± 2.7 [c] | 17.8 ± 2.5 [b] | 12.8 ± 0.4 [a] | <0.001 |
| *h* | 81.5 ± 2.3 [c] | 68.9 ± 3.1 [b] | 25.0 ± 4.4 [a] | <0.001 |
| Chemical Parameters: | | | | |
| Total fat (%, d.m.) | 60.2 ± 2.6 [a] | 64.3 ± 2.3 [b] | 67.3 ± 1.9 [b] | <0.001 |
| Protein (%, d.m.) | 4.77 ± 0.35 [b] | 4.30 ± 0.37 [a,b] | 3.78 ± 0.15 [a] | 0.020 |
| Ashes (%, d.m.) | 14.99 ± 0.03 [b] | 13.58 ± 0.05 [a] | 13.85 ± 0.01 [a] | <0.001 |
| Carbohydrates (%, d.m.) | 20.0 ± 2.7 [b] | 17.8 ± 2.6 [a,b] | 15.0 ± 1.9 [a] | 0.010 |
| Energetic value (kcal/100 g d.m.) | 641 ± 13 [a] | 667 ± 12 [b] | 681 ± 9 [b] | <0.001 |

Different letters on the same line indicate the existence of significant differences ($p < 0.05$). d.m. = dry matter.

Regarding the chroma (color intensity, evaluated by the parameter *C**), the "green table olive flour" was the one with the highest purity or color intensity, followed by the "turning color", and "black table olive" flours. As expected, the three flours presented different hues (values of *h*), taking into account their visual aspect (Figure 1). These results are in line with those stated by Barros et al. [28] during olive fruits ripening of the cultivars Blanqueta, Cobrançosa, and Galega. These authors also observed a decrease in *L**, *b**, *C** and *h* colorimetric parameters throughout the fruit ripening period.

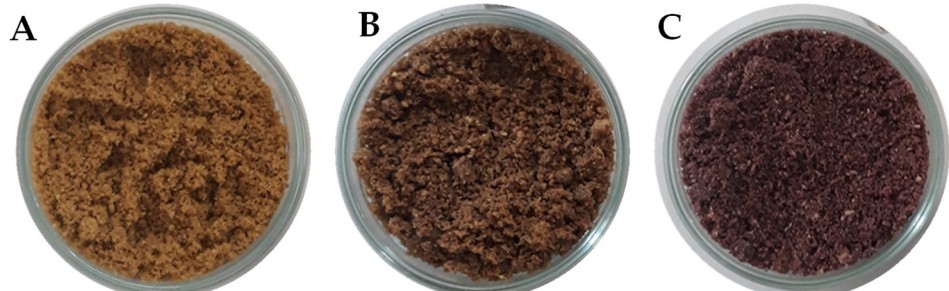

**Figure 1.** Visual appearance of the table olive flours: (**A**) Green; (**B**) Turning color; (**C**) Black.

*3.2. Nutritional Composition*

The moisture contents of the olives (raw material) varied between 70.8 ± 0.3 (turning color olives) and 73.2 ± 0.3 (green olives), with black olives presenting 71.7 ± 3.2% (results not shown). Reduced, but still significant, differences between turning color olives and the remaining ones were present ($p = 0.01$). These values are in line with those expected for freshly cured olives [29].

For the flours produced from table olives at different color types (Table 1), all of them presented moisture contents lower than 2%. The fat content varied between 60.2 and 67.3% (d.m.), with the lowest value corresponding to the "green table olive flour", and no significant differences were detected between the flours obtained from turning color and black table olives. On the contrary, for the crude protein content, the highest value was determined in the "green table olive flour" (4.77%, d.m.), decreasing for the "turning color" and "black table olive" flours (4.30 and 3.78%, d.m., respectively). As far as carbohydrates are concerned, their contents varied between 15.0% (d.m.) for "black table olive flour" and 20.0% (d.m.) for "green table olive flour". These results show that as the olives ripen, the crude protein and carbohydrate contents decrease due to oil increase.

Finally, regarding the ash content, the "green table olive flour" had the highest value (14.99%, d.m.), while the remaining flours showed contents of 13.58 and 13.85% (d.m.) ("turning color", and "black table olive" flours, respectively).

Globally, the energy values varied between 641 and 681 kcal/100 g (d.m.), with the lowest value being found in "green table olive flour" and the highest in "black table olive flour", not significantly different from that determined in "turning color table olive flour". These results are because these two "flours" are the ones that presented the highest fat contents.

### 3.3. Fatty Acids Profile

Table 2 shows the fatty acid composition of the extracted lipids from the three types of "table olive flours" from cv. Cobrançosa, together with the amount of tocopherols present.

**Table 2.** Fatty acid profile (relative%) and tocopherols (g/100 g d.m.) determined in table olive flours.

| Fatty Acid | Table Olive Flour | | | |
|---|---|---|---|---|
| | **Green** | **Turning Colour** | **Black** | ***p*-Value** |
| C16:0 | 11.88 ± 0.04 [a] | 12.39 ± 0.12 [b] | 12.82 ± 0.19 [c] | <0.001 |
| C17:0 | 0.21 ± 0.01 | 0.22 ± 0.01 | 0.20 ± 0.01 | 0.194 |
| C18:0 | 3.97 ± 0.12 [a] | 4.62 ± 0.15 [b] | 4.57 ± 0.31 [b] | <0.001 |
| C20:0 | 0.44 ± 0.01 | 0.45 ± 0.01 | 0.41 ± 0.03 | 0.084 |
| C22:0 | 0.12 ± 0.01 [c] | 0.11 ± 0.01 [b] | 0.10 ± 0.01 [a] | 0.002 |
| C24:0 | 0.07 ± 0.01 [c] | 0.06 ± 0.01 [b] | 0.05 ± 0.01 [a] | <0.001 |
| SFA | 16.74 ± 0.15 [a] | 17.89 ± 0.23 [b] | 18.20 ± 0.19 [c] | <0.001 |
| C16:1 | 0.92 ± 0.01 [a] | 1.03 ± 0.01 [b] | 1.21 ± 0.05 [c] | <0.001 |
| C17:1 | 0.26 ± 0.01 [c] | 0.25 ± 0.01 [b] | 0.24 ± 0.01 [a] | <0.001 |
| C18:1 | 71.89 ± 0.29 [c] | 68.20 ± 0.39 [b] | 66.20 ± 0.35 [a] | <0.001 |
| C20:1 | 0.47 ± 0.01 [b] | 0.36 ± 0.07 [a.b] | 0.31 ± 0.14 [a] | 0.042 |
| MUFA | 73.55 ± 0.30 [c] | 69.84 ± 0.34 [b] | 67.96 ± 0.34 [a] | <0.001 |
| C18:2 | 6.31 ± 0.05 [a] | 8.56 ± 0.32 [b] | 10.24 ± 0.40 [c] | <0.001 |
| C18:3 | 1.37 ± 0.02 [c] | 1.25 ± 0.02 [b] | 1.15 ± 0.03 [a] | <0.001 |
| C22:2 | 0.76 ± 0.04 [b] | 0.68 ± 0.03 [a] | 0.70 ± 0.02 [a] | 0.001 |
| PUFA | 8.43 ± 0.07 [a] | 10.50 ± 0.32 [b] | 12.10 ± 0.39 [c] | <0.001 |
| α-tocopherol | 19.0 ± 0.9 [b] | 17.5 ± 1.0 [b] | 12.5 ± 2.8 [a] | 0.001 |
| β-tocopherol | 0.31 ± 0.01 [a] | 0.47 ± 0.02 [b] | 0.54 ± 0.07 [b] | <0.001 |
| γ-tocopherol | 0.96 ± 0.07 [c] | 0.66 ± 0.03 [b] | 0.54 ± 0.03 [a] | <0.001 |
| Total | 20.3 ± 1.0 [b] | 18.6 ± 1.0 [b] | 13.6 ± 2.8 [a] | 0.001 |

Different letters on the same line indicate the existence of significant differences ($p < 0.05$).

In line with olive oil composition, the major fatty acid was oleic acid (C18:1), with percentages ranging between 66.20 and 71.89%, supporting the high percentages of monounsaturated fatty acids (MUFA). The highest percentages of oleic acid and MUFA were determined in the "green table olive flour" (71.89 and 73.55%, respectively), while the lowest values were obtained in the "black table olive flour" (66.20 and 67.96%, respectively).

Palmitic acid (C16:0), the second fatty acid, varied between 11.88 and 12.82%, and was also the main saturated fatty acid in a total of 16.74 to 18.20% of saturated fatty acids (SFA). The highest relative percentage for SFA corresponded to the "black table olive flour", and the lowest to the "green table olive flour".

Regarding polyunsaturated fatty acids (PUFA), the amounts varied between 8.4 and 12.1%, with linoleic acid (C18:2) being the main component, with values ranging between 6.31 and 10.24%. The linolenic acid (C18:3) presented percentages between 1.15 and 1.37%. For PUFA, the highest percentage corresponded to the "black table olive flour" and the lowest value to the "green table olive flour".

### 3.4. Tocopherol Profile

Table 2 shows the tocopherols determined in the "table olive flours" (green, turning color, and black). In total, three tocopherols were detected in the "flours", in the proportion $\alpha$-tocopherol > $\gamma$-tocopherol > $\beta$-tocopherol. The "green table olive flour" had the highest values of $\alpha$-tocopherol (19.0 ± 0.9 g/100 g, d.m.) and $\gamma$-tocopherol (0.96 ± 0.07 g/100 g, d.m.), compared to the values obtained for the other two "flours". However, the "turning color table olive flour" had an $\alpha$-tocopherol content (17.5 ± 1.0 g/100 g, d.m.) which was not significantly different to the value determined in the "green table olive flour". On the contrary, the "green table olive flour" had the lowest $\beta$-tocopherol value (0.31 ± 0.01 g/100 g, d.m.). This tocopherol was present in greater quantity in the "black" and "turning color table olive flours" (0.54 ± 0.07 and 0.47 ± 0.02 g/100 g, d.m., respectively).

Therefore, and as shown in Table 2, it can be concluded that the "green table olive flour" was the one with the highest tocopherol content (20.3 ± 1.0 g/100 g, d.m.), followed by the "turning color table olive flour" (18.6 ± 1.0 g/100 g, d.m.), and therefore with a higher content of vitamin E and antioxidant activity derived from this compound. On the contrary, the "black table olive flour" was that with the lowest value of tocopherols (13.6 ± 2.8 g/100 g, d.m.), including $\alpha$- and $\gamma$-tocopherols.

### 3.5. Phenolic Composition

The phenolic composition of "table olive flours from cv. Cobrançosa" was determined by HPLC-DAD. In Figure 2, three chromatographs of the "green, turning color, and black table olive flours" are presented.

In total, nine compounds were identified and quantified, namely: hydroxytyrosol, tyrosol, oleuropein derivatives, chlorogenic acid, verbascoside and one derivative, rutin, luteolin, and apigenin. Table 3 shows the contents of these different compounds in the three "table olive flours". The total contents varied between 247 and 404 mg/100 g d.m.

Of all the compounds identified, hydroxytyrosol was the major compound in all "table olive flours", varying between 102 and 223 mg/100 g d.m., suggesting that table olives are generally good sources of this phenol. The "green table olive flour" was the one with the highest amount of this compound, followed by the "turning color table olive flour" and, finally, the "black table olive flour".

**Table 3.** Phenolic profile (mg/100 g dry matter), total reducing capacity (mg GAE/g extract) and radical scavenging activity of DPPH and ABTS of different table olive flours.

| Phenolic Compounds | Table Olive Flour | | | *p*-Value |
|---|---|---|---|---|
| | Green | Turning Colour | Black | |
| Hydroxytyrosol | 223 ± 11 [c] | 160 ± 27 [b] | 102 ± 4 [a] | <0.001 |
| Tyrosol | 83 ± 2 [b] | 56 ± 12 [a] | 44 ± 6 [a] | <0.001 |
| Chlorogenic acid | ND | 2.1 ± 0.4 [a] | 3.1 ± 0.4 [b] | 0.001 |
| Oleuropein derivatives | 31 ± 3 | ND | ND | - |
| Verbascoside derivatives | ND | 10 ± 3 [a] | 12 ± 2 [a] | 0.126 |
| Verbascoside | ND | 20 ± 6 [a] | 27 ± 5 [a] | 0.080 |
| Rutin | 12 ± 3 [a] | 11 ± 2 [a] | 13 ± 1 [a] | 0.108 |
| Luteolin | 49 ± 1 [b] | 46 ± 2 [a,b] | 42 ± 3 [a] | 0.003 |
| Apigenin | 5.4 ± 0.2 [b] | 3.9 ± 0.2 [a] | 4.1 ± 0.3 [a] | <0.001 |
| Total | 404 ± 16 [c] | 309 ± 32 [b] | 247 ± 19 [a] | <0.001 |
| Antioxidant activity | | | | |
| TRC (mg GAE/g extract) | 247.3 ± 14.2 [a] | 302.0 ± 41.3 [b] | 354.4 ± 13.9 [c] | <0.001 |
| RSA DPPH (% inibition) | 90.5 ± 0.5 [b] | 84.8 ± 6.7 [a,b] | 78.1 ± 10.2 [a] | 0.005 |
| RSA ABTS$^{\bullet+}$ (% inibition) | 54.8 ± 1.8 [c] | 47.0 ± 7.5 [b] | 39.3 ± 4.7 [a] | <0.001 |

Different letters on the same line indicate the existence of significant differences ($p < 0.05$).

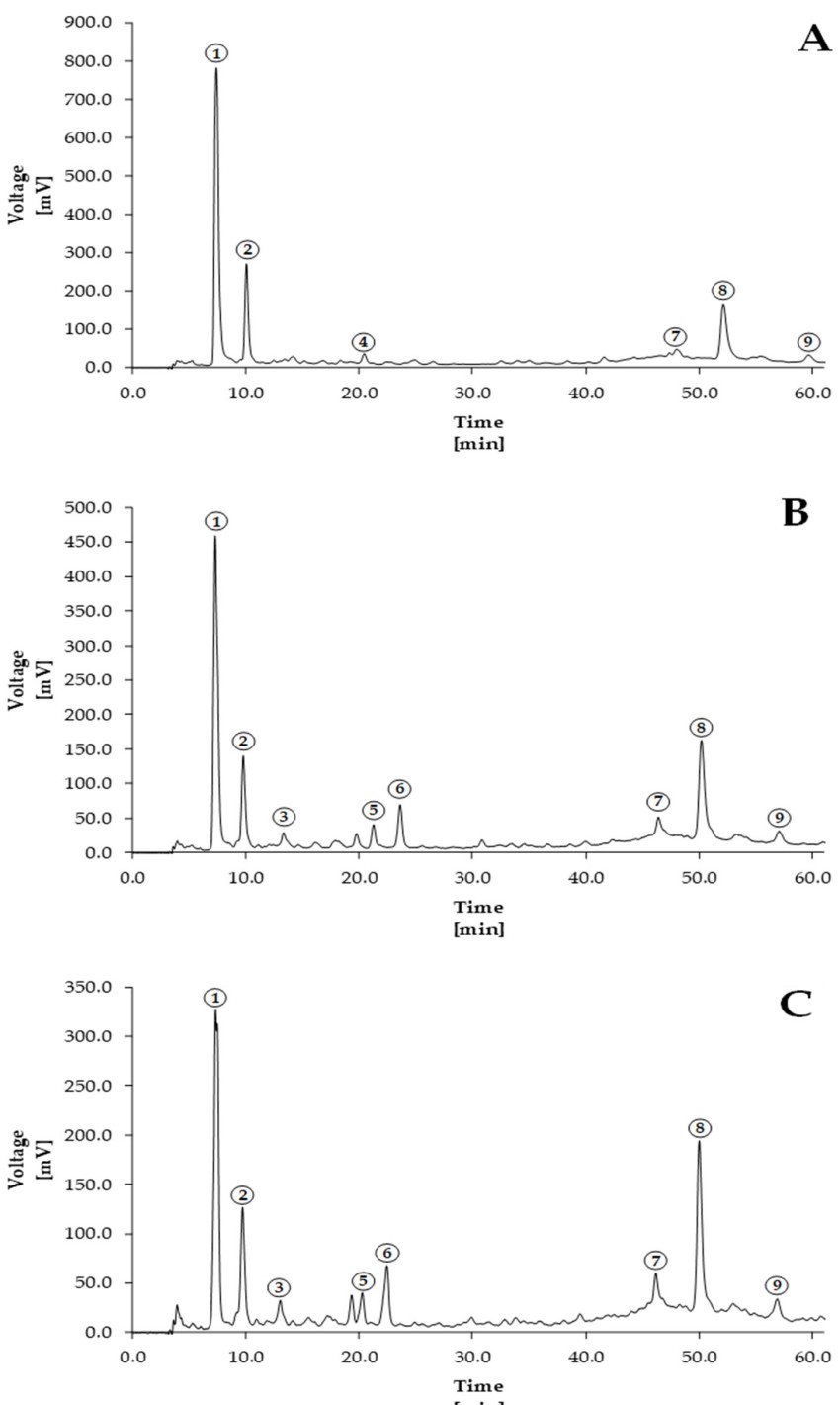

**Figure 2.** Chromatographic profile of the different table olive flours (**A**) Green; (**B**) Turning color; (**C**) Black, obtained by HPLC-DAD. The numbers correspond to the following compounds identified: 1—Hydroxytyrosol; 2—Tyrosol; 3—Chlorogenic acid; 4—Oleuropein derivative; 5—Verbascoside derivative; 6—Verbascoside; 7—Rutin; 8—Luteolin; 9—Apigenin.

The second major compound observed was tyrosol, ranging from 44 to 83 mg/100 g d.m. The highest value was again obtained in the "green table olive flour". No significant differences were detected among the "turning color" and "black table olive" flours.

As the third major compound in the different flours, luteolin stood out with a concentration between 42 and 49 mg/100 g d.m. The "green" and "turning color table olive" flours showed the highest values. On the contrary, oleuropein was not detected.

Rutin and apigenin were the compounds detected in smaller quantities in the "table olive flours". No significant differences were observed between them concerning rutin. However, a higher apigenin value was observed in the "green table olive flour" compared to the other two flours.

### 3.6. Antioxidant Activity

The antioxidant activity of the different "table olive flours" was evaluated through the total reducing capacity and the radical scavenging effects of DPPH and ABTS$^{\bullet+}$ free radicals (Table 3).

Regarding the total reducing capacity (TRC), it varied between 247.3 mg equivalent of gallic acid/g of extract for "green table olive flour" and 354.4 mg equivalent of gallic acid/g of extract for the "black table olive flour" (Table 3).

Free radical scavenging is one of the known mechanisms by which antioxidants inhibit lipid oxidation. Thus, it is of particular importance in lipids. Concerning the blocking effect of DPPH free radicals, the values varied between 78.1 and 90.5% of inhibition, with the highest values being obtained in "green table olive flour".

The same trend was observed for the ABTS$^{\bullet+}$ free radical scavenging activity (Table 3). The inhibition percentages varied between 39.3 and 54.8%, with the "green table olive flour" showing the highest sequestering activity again. The "black table olive flour" showed the lowest antioxidant activity.

### 3.7. Discrimination of the "Table Olive Flours" Based on the Physicochemical and Antioxidant Properties

To summarize the data obtained in the physicochemical and antioxidant properties of the three "table olive flours", a principal component analysis (PCA) was performed. Overall, 87.4% of the total variance of the data could be explained using two principal factors (PC1 = 73.0%; PC2 = 14.4%) (Figure 3). Samples were naturally gathered into three main groups: Group I is represented in the negative region of the first principal factor ("green table olive flour"); Group II is represented in the central region of the figure in the positive region of the second principal factor ("turning color table olive flour"), and Group III represented in the positive region of the first principal factor ("black table olive flour").

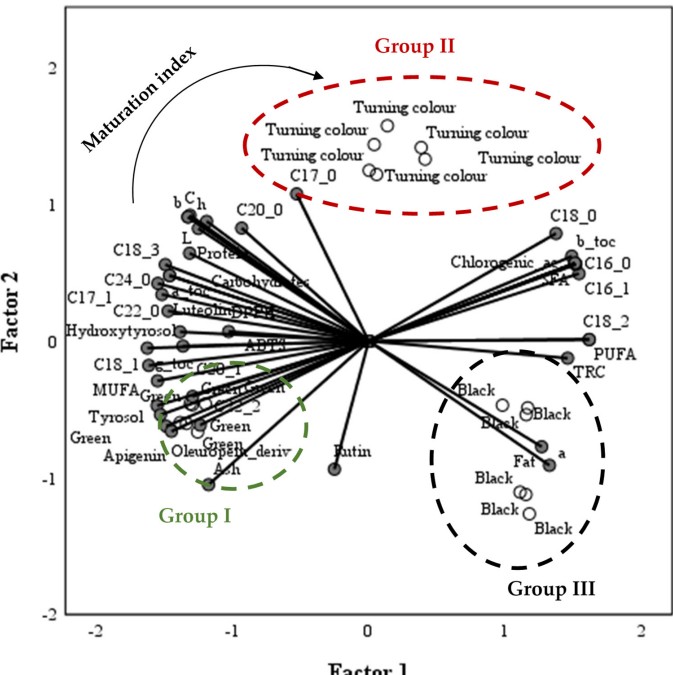

**Figure 3.** Principal components analysis obtained from the physic-chemical and antioxidant properties of "table olive flours" obtained from naturally fermented olives at different maturation indexes. The PCA factors explained 87.4% of the total variance.

## 4. Discussion

In general, flours with distinct colors and nutritional characteristics were obtained when different natural table olives were used. Concerning the fat content of the flours, as mentioned before, the lowest value corresponded to the "green table olive flour", and no significant differences were detected between the flours obtained from turning color and black table olives. This fact is related to the maturation process of the fruits. During this process, some biochemical changes occurred, and the amount of fat increased [30]. Identical results were mentioned by Ünal & Nergiz [31], who indicated a lower fat content in green olives compared to black olives. On the contrary, the crude protein and carbohydrate contents decrease as the olives ripen due to this oil increase. Regarding the ash content, the "green table olive flour" had the highest value. This mineral content is expected to be mostly due to the salt used for olive curing. Since green table olives presented higher moisture and the brine used always had the same salt concentration, the amount of salt will be higher after removing the water to express the results in dry weight. Consequently, the ash will be higher in the green table olives.

Regarding fatty acids and tocopherols, the olives' ripeness influences the profile and content of these compounds. The "green table olive flour" had the highest percentages of oleic acid (C18:1) and MUFA. On the contrary, the "black table olive flour" had the highest percentages of linoleic acid (C18:2) and PUFA, and the more mature the olives are, the fewer tocopherols their flours will have.

The total phenolic contents varied between 247 and 404 mg/100 g d.m., in line with the values reported by Pereira et al. [4] (87–460 mg/100 g d.m.) for Portuguese table olives. These results are similar to those reported by Sousa et al. [11], who also detected a decline in total phenolics during the maturation process of cv. Cobrançosa. As mentioned before, of all the compounds identified, hydroxytyrosol was the major compound in all "table olive flours", suggesting that table olives are generally good sources of this phenol. Similar results for table olives were reported by Romero et al. [32], Blekas et al. [8] and Pereira et al. [4]. According to these authors, hydroxytyrosol is the phenolic compound that exists in more significant quantities in table olives. Its accumulation comes from the hydrolysis of oleuropein and hydroxytyrosol-4-β-glucoside, phenolic compounds that decreased during the fermentation and storage time [32]. Hydroxytyrosol is a compound with antioxidant and antimicrobial properties, and is already recognized by the FDA as a safe compound, designated as GRAS (Generally Recognized as Safe) [33]. The "green table olive flour" was the one with the highest amounts of this compound, followed by the "turning color table olive flour" and, finally, the "black table olive flour". Concerning tyrosol, similar to hydroxytyrosol, the amount of both compounds decreases as the olive matures. Tyrosol was also detected by Romero et al. [32] when studying the fermentation of naturally black olives. Tyrosol also has antioxidant activity, and is an anti-arrhythmia agent [33]. Luteolin stood out as the third major compound, with the "green and turning color table olive flours" showing the highest values. Luteolin is a flavonoid, with antioxidant and anti-inflammatory potential, as well as apoptosis-inducing and chemo preventive activities [33].

Oleuropein was not detected since, during fermentation, it is known that this phenol decreases with time, being converted in hydroxytyrosol [32]. Oleuropein is responsible for the bitter taste of unprocessed olives and, to become edible, the fruits need to lose, at least partially, their natural bitterness. Consequently, it is expected that oleuropein might not be found in processed fruits. Blekas et al. [8] also did not find oleuropein when studying 25 samples of tables olives (Spanish-style green olives in brine, Greek-style naturally black olives in brine, and Kalamata olives in brine) purchased in Greece, as well as Pereira et al. [4] when studying Portuguese table olives. However, as can be seen by observing Table 3, it was found that an oleuropein derivative was detected in the "green table olive flour", neither being seen in the "turning color table olive flour" nor in the "black table olive flour". On the contrary, chlorogenic acid, verbascoside, and one of its derivatives were only detected and quantified in the "turning color" and "black table olive flours". In the "black table olive flour", they were found in more significant quantities.

However, it was only in chlorogenic acid that a significant difference was observed between the two flours. Verbascoside was also only quantified by Sousa et al. [11] in mature fruits of cv. Cobrançosa, while in immature fruits, this compound was not detected, in line with our results. Ryan and Robards [7] suggested that the formation of verbascoside may also be related to the partial degradation of oleuropein, which could explain the later appearance of this compound in olive fruits. Furthermore, Ferro et al. [30] also referred that verbascoside formation is metabolically linked to the conjugation of hydroxytyrosol with caffeic acid. In the present work, it was stated that when the concentration of hydroxytyrosol decreased with the ripening stage, the verbascoside increased.

Rutin and apigenin were the compounds detected in smaller quantities. Nevertheless, both compounds have interesting health properties. Rutin is a vasoprotective, and apigenin has a biological activity to inhibit tumor growth, as well as chemopreventive activity [33].

In general, regarding the sum of identified phenolic compounds, the sequence obtained was as follows: "green table olive flour"> "turning color table olive flour"> "black table olive flour". However, all of the "olive flours" produced in the present work showed phenolic compounds with biological properties that are very important for human health.

Regarding the antioxidant activity, it was stated that the total reducing capacity increased as the olive ripeness increased. These results may be due to the fact that this test evaluates the presence of compounds with reducing ability, such as sugars, organic acids or peptides, and is not only specific for phenolic compounds. Other substances may be present in the table olives, which may have reducing properties. Concerning the blocking effect of DPPH free radicals, the highest values were obtained in "green table olive flour". Thus, it was found that as maturation increases, the blocking effect of DPPH free radicals decreases, indicating that the antioxidant capacity assessed by the sequestration of these free radicals was more significant in lower ripening states, that is, in the "green table olive flour". This is in line with Sousa et al. [11], who also indicated a lower antioxidant potential evaluated by the DPPH method with increased maturity in olives of cv. Cobrançosa. The same trend was observed for the ABTS$^{\bullet+}$ free radical scavenging activity. These results may also be related to the lower content of Vitamin E (tocopherols) in the "black table olive flour", as Vitamin E is recognized as a powerful antioxidant in olives. Vitamin E has an important role in the preservation of lipid moiety.

After performing a PCA, the "table olive flours" were discriminated into three groups based on the physicochemical and antioxidant properties. Group I ("green table olive flour") corresponded to the samples with higher C18:1 and MUFA proportions, and with the highest DPPH and ABTS$^{\bullet+}$ free radical scavenging activities, corresponding to a high antioxidant activity. This fact was probably due to the higher contents in hydroxytyrosol, tyrosol and γ-tocopherol. Group II ("turning color table olive flour") included the samples with one of the highest C18:0 percentage. This group is located in the middle of the other two groups. Finally, Group III ("black table olive flour") corresponded to the samples with higher *a*\*, C18:2, C16:1, PUFA, and SFA proportions, as well as the highest total reducing capacity.

## 5. Conclusions

"Table olive flours" with different colors and different characteristics were obtained. The "green table olive flour" had the lowest fat content and energy value, in comparison with the other two flours. On the contrary, it showed higher protein and carbohydrate contents than "black olive flour". The "green olive flour" was the one with the highest percentages of oleic acid (C18:1) and MUFA, as well as γ-tocopherol. It also showed the highest antioxidant activity, assessed through the radical scavenging activities of DPPH and ABTS$^{\bullet+}$. The "black table olive flour" was the one with the highest percentages of linoleic acid (C18:2) and PUFA, as well as the highest total reducing capacity. In the three "flours" developed, nine phenolic compounds were detected, with hydroxytyrosol being the major, followed by tyrosol, and luteolin. We can conclude that it is possible to produce different table olives flours with distinct properties. This will allow meeting new consumers and

trends in the search for differentiated clean-label products. Furthermore, the production of this new product will also reduce waste and be more sustainable, in line with the objectives of the circular economy. Due to their composition, it is also expected that the flours prepared with green olives could potentially have a longer shelf life, deserving further experiments to confirm it and the acceptability of consumers when included in recipes.

**Author Contributions:** J.A.P. planned and supervised the work, revised the final written work and was responsible for funding acquisition; C.O. executed the laboratory work; N.R. and E.R. wrote the manuscript and treated the results; S.C. performed the characterization of the lipid fraction. All authors have read and agreed to the published version of the manuscript.

**Funding:** This research was funded by Foundation for Science and Technology (FCT, Portugal). The authors are grateful for the financial support by national funds FCT/MCTES to CIMO (UIDB/00690/2020), SusTEC (LA/P/0007/2020) and REQUIMTE-LAQV (UIDB/50006/2020) units, AgriFood XXI I&D&I project (NORTE-01-0145-FEDER-000041) and GreenHealth Project (NORTE-01-0145-FEDER-000042) co-financed by European Regional Development Fund (ERDF) through the NORTE 2020 (Programa Operacional Regional do Norte 2014/2020). Nuno Rodrigues thanks National funding by FCT—Foundation for Science and Technology, P.I., through the institutional scientific employment program contract.

**Institutional Review Board Statement:** Not applicable.

**Informed Consent Statement:** Not applicable.

**Data Availability Statement:** Not applicable.

**Conflicts of Interest:** The authors declare no conflict of interest.

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
