# Peer review of "“Table Olive Flours”: An Ingredient Rich in Bioactive Compounds?"

_applsci, doi:10.3390/app12031661_

Round 1

Reviewer 1 Report

General comment

This work focuses on the production and characterization of flours obtained from 3 table olive types. The manuscript is well written in terms of English, and inclusion of references. My main concern is related to the lack of an experimental design in this study. The authors took 3 types of olives, dried and characterized them, but there is not a scientific objective behind these experiments. This is the reason why the PCA analysis cannot be interpreted since there is no real link between the characteristics included in the analysis. I would have expected to find, for example, a storage experiment, that would have certainly impacted the characteristics measured in this study (e.g. antioxidant capacity).

Specific comments

Title: An ingredient rich in bioactive compounds?

L36-37: This sentence is vague, which problems? Which needs?

L37-38: Can table olives be used as meat replacers? According to your results, they contain less than 5% of protein. Moreover, the authors highlighted from the beginning the bioactives content. Therefore, this sentence should be modified or removed.

L42-54: Here, many aspects of olives are described, going from nutritional aspects towards processing and even marketing and sustainability. However, there is not a clear point of discussion on the importance of producing olive flours.

L64-65: Dehydration can decrease perishability by microorganisms but lipid oxidation can be significantly accelerated in dried products, especially in olives containing high oil concentrations.

L66-67: Needs grammar check.

L88: Specify brine composition (NaCl).

L92: So water was changed only once? If so, modify it because now it seems you have changed it several times.

L499-508: I do not completely understand the interpretation of PCA analysis. None of the oil flours contained higher concentrations C17, C18, etc. It seems the authors just listed the common characteristics they obtained but none of these characteristics are really related. This is because there was no experimental design, so no treatments (e.g. different processing levels, or storage time).

Author Response

Reviewer #1

We would like to thank the reviewer effort and dedicated time to evaluate our manuscript. The points addressed by the reviewer allowed us to improve our work considerably.

This work focuses on the production and characterization of flours obtained from 3 table olive types. The manuscript is well written in terms of English, and inclusion of references. My main concern is related to the lack of an experimental design in this study. The authors took 3 types of olives, dried and characterized them, but there is not a scientific objective behind these experiments. This is the reason why the PCA analysis cannot be interpreted since there is no real link between the characteristics included in the analysis. I would have expected to find, for example, a storage experiment, that would have certainly impacted the characteristics measured in this study (e.g. antioxidant capacity).

Answer: Thanks for your comments. We answered these points in the following sections. Also, when applied, we incorporated your improvement in the manuscript.

Specific comments

Title: An ingredient rich in bioactive compounds?

Answer: Thanks for the suggestion. In the revised version, the title was changed according to your suggestion to ““Table olive flours”: an ingredient rich in bioactive compounds?”.

L36-37: This sentence is vague, which problems? Which needs?

Answer: When the sentence “There is also a concern to solve nutritional problems or to meet some specific needs of consumers” was included in the introduction section, we intended to transmit the idea that in the modern style of life, without time to spend for cooking or to search the best ingredients, due to bad feeding habits, in some situations exist nutritional issues, such as dietary imbalances (lack of specific nutrients, antioxidants and vitamins). Furthermore, the existence of food ingredients/products that are more practical, easy-to-use and nutritionally balanced responds to the needs of consumers. At the same time, there is a lot of waste of products that should be used. For example, due to quality aspects, such as fruits with no commercial size or containing some visual defects are wasted. Moreover, the consumers have specific needs, such as promoting the circular economy principles to look forward to the zero-waste objective and reduce the carbon footprint.

In this context, and to solve some possible redundancy, the sentence was changed to:

There is a concern to solve nutritional issues such as dietary imbalances (lack of specific nutrients, antioxidants and vitamins), and the needs of consumers of new ingredients, practical and easy-to-use that also meet environmental and sustainability concerns.

L37-38: Can table olives be used as meat replacers? According to your results, they contain less than 5% of protein. Moreover, the authors highlighted from the beginning the bioactives content. Therefore, this sentence should be modified or removed.

Answer: Thanks for your suggestion. We changed the sentence to “Thus, more innovations are needed, particularly in the direction of plant-based products that are healthy and towards the use of clean labels [2].” (Text added in the Introduction section)

L42-54: Here, many aspects of olives are described, going from nutritional aspects towards processing and even marketing and sustainability. However, there is not a clear point of discussion on the importance of producing olive flours.

Answer: We agree with the reviewer. So, we added the following sentence: “The production of “table olive flour” can be an interesting method to obtain a product that can be added to other dishes or products (such as sauces) to enrich their bioactive properties. Furthermore, a product with lower salt content than the fresh olives can be obtained because before making the flour, the fruits can be soaked to remove a large part of the salt present in the fruit.” (Text added in the Introduction section)

L64-65: Dehydration can decrease perishability by microorganisms but lipid oxidation can be significantly accelerated in dried products, especially in olives containing high oil concentrations.

Answer: We totally agree with the reviewer. In the revised version of the manuscript, the sentence was removed.

L66-67: Needs grammar check.

Answer: The sentence was changed to “Temperatures at 40 [19], 45 [14], and 50 °C [13,15,18], as well as ranges between 40-70 °C [16,17], have been applied;”

L88: Specify brine composition (NaCl).

Answer: In the revised version, the brine composition was specified. Please see the Materials and Methods Section.

L92: So water was changed only once? If so, modify it because now it seems you have changed it several times.

Answer: This aspect was clarified in the manuscript, and the sentence was changed to “To desalt the table olives in the laboratory, fruits were put in distilled water for 48 hours, being the water changed only once after 24 hours.”.

L499-508: I do not completely understand the interpretation of PCA analysis. None of the oil flours contained higher concentrations C17, C18, etc. It seems the authors just listed the common characteristics they obtained but none of these characteristics are really related. This is because there was no experimental design, so no treatments (e.g. different processing levels, or storage time).

Answer: Thanks for your remark and the suggestion for future works. There was a mistake in the text of that section. The sentence was changed to “Group II (“turning colour table olive flour”) included the samples with one of the highest C18:0 percentage. This group is located in the middle of the other two groups.”.

We want to explain that only “flours” obtained from fruits at different ripeness indexes were compared in the present work, but all the fruits followed the same processing method. Thus, we did not want to introduce a new variable linked to the production process. Moreover, we also did not want to study the effect of storage time at this stage. We consider that these aspects will be of high interest in future studies.

Reviewer 2 Report

After reviewing the manuscript “Table olive flours”: a rich ingredient in bioactive compoundsl”, the manuscript presents a well structure and correct presentation of results.

English language and style a little require editing.

Pretty well-described methodology.
However, I have doubts whether the color in the form as it is can be a reference to the comparison?

The other results seem to be correct and interesting. Extensive for information.
Correct discussion.
Current and correct literature

Author Response

Reviewer #2

We would like to thanks for the reviewer effort and dedicated time to evaluate our manuscript. The points addressed by reviewer allowed us to improve considerably our work.

After reviewing the manuscript “Table olive flours”: a rich ingredient in bioactive compounds”, the manuscript presents a well structure and correct presentation of results.

English language and style a little require editing.

Answer: Thank you for your comment. The revised version of the manuscript was corrected and revised by an English native speaker.

Pretty well-described methodology. However, I have doubts whether the colour in the form as it is can be a reference to the comparison?

Answer: In the table olives, the maturation index is defined according to the skin colour and pulp of the fruits. Furthermore, it is also known that the maturation index influences the composition of the fruits and their bioactive properties. In this context, the colour of the fruits was used to find out relationships between the maturation index and table olive flour composition and antioxidant activity.

The other results seem to be correct and interesting. Extensive for information. Correct discussion. Current and correct literature

Answer: Thanks for your comments.

Reviewer 3 Report

Dear Authors,

Manuscript ID: applsci-1583872

This manuscript reports about "Table olive flour"  as an innovative product that can gain importance as a bioactive rich ingredient. The Authors explored the possibilities of using these fruits to flour production.

 The paper presents a study that is incredibly relevant in the light of arguments used to support it. Research is very gripping as well as it also  has its  scientific value. I think the motivations for this study are very clear. The introduction provides a good, generalized background of the topic that quickly gives the reader an  appreciation of the scientific relevance and timeliness of the research theme. Manuscript is well-written.

The research has been properly planned.  The experimental methods are appropriate for the study and  the data didn’t duplicate in the graphics and text. In my opinion the findings of this manuscript are also properly described in the context of the published literature.

It should be noted that the obtained results of the study have an extremely high application potential. It was a great pleasure to read the text of this manuscript.

However, there are minor flaws of the manuscript that  need to be fixed before publication.

Specific comments on the manuscript are as follows:

  • Please, clearly indicate in one sentence in Abstract:

“The aim of this study was…”

  • Line 30: Please complete 1 keyword which describes analyzed products.
  • In the subsection 2.6, please complete information about PCA analysis:

How many principal components finally were identified and how many % of the total variance were explained ( data presented under Figure 3)?

  • In the subsection 2.6, please complete information about statistical program (manufacturer, town, country).
  • About half of the cited literature dates back 10-15 years. Perhaps Authors could make some changes in the manuscript regarding this issue.
  • Here are a few current references to consider, authors can use them in the article:
  1. Amanda Vaccalluzzo, Alessandra Pino, Nunziatina Russo, Maria De Angelis, Cinzia Caggia, Cinzia Lucia Randazzo, FoodOmics as a new frontier to reveal microbial community and metabolic processes occurring on table olives fermentation, Food Microbiology, Volume 92, 2020, 103606, ISSN 0740-0020, https://doi.org/10.1016/j.fm.2020.103606.

(https://www.sciencedirect.com/science/article/pii/S0740002020301957)

  1. William Leonard, Pangzhen Zhang, Danyang Ying, Benu Adhikari, Zhongxiang Fang,

Fermentation transforms the phenolic profiles and bioactivities of plant-based foods, Biotechnology Advances, Volume 49, 2021, 107763, ISSN 0734-9750,

https://doi.org/10.1016/j.biotechadv.2021.107763.

(https://www.sciencedirect.com/science/article/pii/S0734975021000690)

  1. Diva Santos, José A. Lopes da Silva, Manuela Pintado, Fruit and vegetable by-products' flours as ingredients: A review on production process, health benefits and technological functionalities, LWT, Volume 154, 2022, 112707, ISSN 0023-6438,

https://doi.org/10.1016/j.lwt.2021.112707. (https://www.sciencedirect.com/science/article/pii/S0023643821018600)

In my opinion, presented the manuscript is appropriate for publication Journal Applied – Sciences, after minor revision, given the above aspects.

Author Response

Reviewer #3

We would like to thanks for the reviewer effort and dedicated time to evaluate our manuscript. The points addressed by reviewer allowed us to improve considerably our work.

This manuscript reports about "Table olive flour" as an innovative product that can gain importance as a bioactive rich ingredient. The Authors explored the possibilities of using these fruits to flour production. The paper presents a study that is incredibly relevant in the light of arguments used to support it. Research is very gripping as well as it also has its scientific value. I think the motivations for this study are very clear. The introduction provides a good, generalized background of the topic that quickly gives the reader an appreciation of the scientific relevance and timeliness of the research theme. Manuscript is well-written. The research has been properly planned.  The experimental methods are appropriate for the study and the data didn’t duplicate in the graphics and text. In my opinion the findings of this manuscript are also properly described in the context of the published literature. It should be noted that the obtained results of the study have an extremely high application potential. It was a great pleasure to read the text of this manuscript. However, there are minor flaws of the manuscript that need to be fixed before publication.

Answer: Thanks for your opinion. It encourages us to continue in the research line of olive products, valorization and innovation.

Specific comments on the manuscript are as follows:

Please, clearly indicate in one sentence in Abstract:

“The aim of this study was…”

Answer: Thanks for your comment. We change to “The aim of this study was to produce different "table olive flours" from fruits at different maturation stages. "Table olive flour" is here presented as an innovative product that can gain importance as a bioactive rich ingredient.” (Please see the Abstract Section in the revised manuscript).

Line 30: Please complete 1 keyword which describes analyzed products.

Answer: We added the word “Table olive flours” to the keywords.

In the subsection 2.6, please complete information about PCA analysis:

Answer: We added the following information: “The PCA score plot was used to differentiate the table olive flours and verify the role of table olives (green, turning colour and black) on their properties, namely, colour, fatty acids and tocopherol compositions, individual phenolic compounds and antioxidant activity. The number of components to keep for data analysis was evaluated by: (i) the respective eigenvalues (must be >1); (ii) Cronbach’s α parameter (that should be positive); and (iii) the total percentage of variance (that should be as high as possible) explained by the number of components selected.”.

How many principal components finally were identified and how many % of the total variance were explained (data presented under Figure 3)?

Answer: Two principal components were identified, explaining 87.4% of the total variance.

In the section entitled “3.7 Discrimination of the "table olive flours" based on the physicochemical and antioxidant properties”, it is indicated that87.4% of the total variance of the data could be explained using two principal factors (PC1 = 73.0%; PC2 = 14.4%) (Figure 3).”.

In the subsection 2.6, please complete information about statistical program (manufacturer, town, country).

Answer: We added the information: “The program used in the statistical analysis was Minitab (version 14, Minitab Ltd., Coventry, United Kingdom).”.

About half of the cited literature dates back 10-15 years. Perhaps Authors could make some changes in the manuscript regarding this issue.

Here are a few current references to consider, authors can use them in the article:

Amanda Vaccalluzzo, Alessandra Pino, Nunziatina Russo, Maria De Angelis, Cinzia Caggia, Cinzia Lucia Randazzo, FoodOmics as a new frontier to reveal microbial community and metabolic processes occurring on table olives fermentation, Food Microbiology, Volume 92, 2020, 103606, ISSN 0740-0020, https://doi.org/10.1016/j.fm.2020.103606.

(https://www.sciencedirect.com/science/article/pii/S0740002020301957)

William Leonard, Pangzhen Zhang, Danyang Ying, Benu Adhikari, Zhongxiang Fang, Fermentation transforms the phenolic profiles and bioactivities of plant-based foods, Biotechnology Advances, Volume 49, 2021, 107763, ISSN 0734-9750,

https://doi.org/10.1016/j.biotechadv.2021.107763.

(https://www.sciencedirect.com/science/article/pii/S0734975021000690)

Answer: We understand the reviewer's point of view, and we thank his/her intention to improve our work. Nevertheless, these articles are focused on the fermentation process, a topic not addressed in the present study. These articles will be really interesting for future works that can be performed on the fermentation of natural table olives.

Diva Santos, José A. Lopes da Silva, Manuela Pintado, Fruit and vegetable by-products' flours as ingredients: A review on production process, health benefits and technological functionalities, LWT, Volume 154, 2022, 112707, ISSN 0023-6438,

https://doi.org/10.1016/j.lwt.2021.112707. (https://www.sciencedirect.com/science/article/pii/S0023643821018600)

Answer: We added this reference in the revised manuscript (Reference [3] included) (Please, check the Introduction Section).

In my opinion, presented the manuscript is appropriate for publication Journal Applied – Sciences, after minor revision, given the above aspects.

Answer: Thanks for your opinion. We tried to follow all the suggestions made by the reviewer that really improved our manuscript.

Round 2

Reviewer 1 Report

Thanks to the authors for addressing my comments.